# Spanish Adaptation of the Overall Anxiety and Depression Severity and Impairment Scales in University Students

**DOI:** 10.3390/ijerph19010345

**Published:** 2021-12-29

**Authors:** Jorge Osma, Víctor Martínez-Loredo, Amanda Díaz-García, Alba Quilez-Orden, Óscar Peris-Baquero

**Affiliations:** 1Departamento de Psicología y Sociología, Universidad de Zaragoza, 44003 Teruel, Spain; loredo@unizar.es (V.M.-L.); amandadiaz@unizar.es (A.D.-G.); 699298@unizar.es (A.Q.-O.); operis@unizar.es (Ó.P.-B.); 2Instituto de Investigación Sanitaria de Aragón, 50009 Zaragoza, Spain; 3Unidad de Salud Mental Moncayo, 50500 Tarazona, Spain

**Keywords:** transdiagnostic, psychometric properties, depression, anxiety, Spanish university students

## Abstract

The lifetime prevalence of emotional disorders in Spain is 4.1% for anxiety and 5.2% for depression, increasing among university students. Considering the scarcity of screenings with adequate psychometric properties, this study aims to explore the validity evidence of the Overall Anxiety/Depression Severity and Impairment Scales (OASIS and ODSIS). A total of 382 university students from the general population were assessed on anxiety and depressive symptoms, as well as quality of life. The one-dimensional structure of both the OASIS and ODSIS explained 87.53% and 90.60% of variance, with excellent internal consistency (α = 0.94 and 0.95, respectively) and optimal cut-offs of 4 and 5, respectively. Both scales show a significant moderate association with other measures of anxiety, depression and quality of life. The OASIS and ODSIS have shown good reliability and sound validity evidence that recommend their use for the assessment and early detection of anxiety and depressive symptoms, and associated quality of life impairment in Spanish youth.

## 1. Introduction

As anxiety, depressive and related disorders entail frequent and intense negative emotions, aversive reactions to the emotional experience and a tendency to dampen, or escape from /avoid it [1], they are referred to as emotional disorders (EDs). In Spain, the lifetime prevalence for anxiety and depressive disorders has been estimated at 4.1% and 5.2%, respectively [2]. EDs are also highly disabling [3] and have been associated with, among others, functional impairment and poorer quality of life [4].

Most psychological problems usually appear at ages 15–24 [5]. The incidence and prevalence of EDs in young people are increasing, with depressive disorders being the dominant health problem [6]. The university period is considered highly stressful because students have to adjust themselves to a different environment and to cope with various stressors such as lifestyle changes [7] that can affect social relationships, academic and work performance [8,9], and quality of life [10]. All these factors may increase the risk for psychological distress [11,12] and mental disorders [7]. Therefore, the university context can have a negative impact on the psychological and emotional well-being of university students [13]. Likewise, it is important to consider that the presence of mental health disorders may be negatively associated and impact some university-related outcomes or academic functioning [14]. Evidence suggests that university students present higher rates of EDs in comparison with non-students in the same age range [15] and adult population [16,17]. Furthermore, EDs among university students are closely associated with other disorders such as substance abuse, personality disorders or suicide attempts [18]. Thus, the early detection of EDs in this population is crucial to shorten the duration of possible episodes of depression and/or anxiety and to avoid functional impairment in the long term [19]. In this regard, the development and validation of rigorous assessment instruments for EDs is an essential task for researchers and clinicians involved in the assessment, prevention and treatment.

Until now, different screening instruments have been designed to assess depression [20,21,22] and anxiety disorders [23,24,25] with good psychometric properties. However, these scales are not designed to capture the severity and impairment of these disorders, regardless of the specific disorder suffered by the patients [26,27], but rather to quantify presence and/or severity of restricted categories of symptoms, under the assumption that the frequency of symptoms relates to the severity of the disorder. In addition, the use of brief instruments to assess multiple anxiety disorders provides a more efficient way to collect data compared to longer scales [28], which may be more useful in public clinical settings.

The Overall Anxiety Severity and Impairment Scale (OASIS [29]) and the Overall Depression Severity and Impairment Scale (ODSIS [30]) were developed to overcome the constraints of the existing tools by (1) assessing transdiagnostic symptoms related to anxiety and depressive symptoms, and multiple domains of clinical severity including functional impairment; (2) assessing the severity of anxiety and depression, regardless of the specific disorder suffered by the patients [26,27]; and (3) be brief (five items each, administered in 2–3 min) and easy to use, which facilitates obtaining relevant data in a more efficient way and allows for the monitoring of symptoms repeatedly throughout a treatment due to its brevity [31].

Both scales are being used increasingly in different countries (OASIS [29,32,33,34,35]) and (ODSIS [30,35,36]) and have shown excellent internal consistency (i.e., between 0.80 and 0.94). Different cut-off points have been found in both scales in clinical samples, the ones found for the OASIS has ranged from 5 to 10 [26,33,37,38,39,40,41] and for the ODSIS, from 5 to 11 [38,42]. The ROC curve analysis has been also used in a recent study in a community sample showing a cut-off point of 15 for the OASIS and 12 for the ODSIS [35]. Among the studies that have included a sample composed of clinical and non-clinical samples or students, the analysis of the ROC curve for both scales has been performed only with the subsample of clinical participants [30,32,36,43]. One study explored the psychometric properties of the OASIS in a sample of students without performing the ROC curve analysis [34]. In Spain, although both questionnaires have been validated in clinical samples in both paper-and-pencil [38] and online version [37,42], their psychometric properties have not yet been examined in nonclinical samples.

The present study investigates the psychometric properties of the Spanish version of the OASIS and ODSIS in a sample of nonclinical university students. Specifically, the main objectives of this study were: (a) to examine the internal structure of both scales; (b) to calculate the internal consistency of both questionnaires; (c) to assess their relationships with other related and unrelated variables; and (d) to obtain cut-off scores to provide appropriate parameters for using the OASIS and ODSIS as screening tools. The precision of both questionnaires was also explored using Item Response Theory (IRT) models.

## 2. Materials and Methods

### 2.1. Participants

From a total sample of *N* = 571 participants currently studying at different Spanish universities, 189 were excluded for not having correctly completed the entire evaluation battery and 4 for being under psychological treatment (see Table 1 for sociodemographic characteristics). The final sample comprised 378 participants (84.90% women; *M*age = 22.71 years old, *SD* = 5.47, range = 18–55).

The following inclusion criteria in the study were followed: (1) being over 18 years of age; (2) being a student at a university in Spain; (3) speaking Spanish fluently, and (4) understanding and accepting the contents of the informed consent, expressed by signing it. There were no additional exclusion criteria.

### 2.2. Instruments

Socio-demographic data. Participants were asked about sex, age, marital status, level of studies achieved, employment situation, university in which they are currently studying, career or studies that they are currently studying, and past and current psychological treatments.

The EuroQol [44,45] is a generic instrument that evaluates health-related quality of life. It has five dimensions (mobility, personal care, daily activities, pain, and anxiety/depression), and each of them has three levels of severity (no problems, some problems or moderate problems and serious problems). Additionally, it has a visual analog scale (VAS) that shows the general state of health perceived through a “thermometer” scale raining from 0 (the worst imaginable state of health) to 100 (the best imaginable health). In the present study, only the VAS was used so Cronbach’s alpha cannot be calculated. We will refer to this variable as VAS-EuroQol.

The Depression, Anxiety and Stress Scale (DASS-21; [46,47]) evaluates dimensions of Stress, Anxiety and Depression. It consists of 21 items whose response format is a Likert-type scale with four response options from 0 (Nothing applicable to me) to 3 (Very applicable to me, or applicable most of the time). Of this scale, the 14 items of the anxiety and depression subscales have been used, so throughout this article we will refer to it as DASS-14. In the present sample, Cronbach’s alpha for the depression subscale was 0.89, and for the Anxiety subscale was 0.82.

The Overall Anxiety Severity and Impairment Scale (OASIS; [29,38]) evaluates severity and impairment associated with anxiety symptoms. It consists of 5 items whose response format is a Likert-type scale with five response options from 0 (I did not feel anxious/little or nothing/none) to 4 (constant anxiety/extreme/all the time).

The Overall Depression Severity and Impairment Scale (ODSIS; [30,38]) assesses severity and impairment in the context of depressive symptoms. It consists of 5 items whose response format is a Likert-type scale with five response options from 0 (I did not feel depressed/little or nothing/none) to 4 (constant depression/extreme/all the time).

### 2.3. Procedure

The study was carried out at the University of Zaragoza using the Qualtrics platform v2.16 (Qualtrics, Provo, UT, USA) [48] for sample collection and application of the evaluation battery. The snowball sampling method was used. The survey was disseminated through instant messaging applications, social networks and advertisements in faculties from different universities in Spain. The message requested participation in the study as well as the dissemination of the message among acquaintances who were also students from different Spanish universities. Participation was voluntary and without financial reward. When the platform link was opened by the participant, the informed consent document appeared and, after approving it, the evaluation battery began (taking approximately 15 min to complete). All the information recruited was anonymous, and alphanumeric codes were used to preserve the data handling. The study has the approval of the Research and Ethics Committee of the Health Research Institute of Aragon (No. CP.-C.I. PI20/053).The Spanish adaptation of the original scales followed the International Test Commission recommendations [49] (see the description of the method at [38]).

### 2.4. Data Analysis

The factorial structure of both questionnaires was examined using exploratory (EFA) and confirmatory (CFA) factor analyses. The total sample was randomly split into two subsamples. The first subsample (*n* = 189) was used to perform the EFAs (unweighted least squares estimator (ULS) extraction method). The optimal implementation of Parallel Analysis [50] (random resampling operations *n* = 1000) was used to determine the optimum number of factors. Goodness of fit were examined through the goodness of fit index (GFI) and the root mean square of the residuals (RMSR) using the following cut-offs: GFI ≥ 0.95, and RMSEA and RMSR ≤ 0.08 [51]. The unidimensional congruence (UniCo), the explained common variance (ECV) and the mean of item residual absolute loadings (MIREAL) were used to examine unidimensionality of both questionnaires using the following cut-offs: UniCo > 0.95, ECV > 0.85 or MIREAL < 0.30 [52]. Cross-validations were performed in the second subsample (*n* = 189) using CFAs with the diagonally weighted least squares (WLSMV) as the estimation method. Goodness of fit was examined via CFI (cut-off ≥ 0.95) and χ^2^/df (cut-off < 5).

The reliability of both questionnaires was estimated using Cronbach’s alpha and McDonald’s omega on the polychoric correlation matrix. Pearson correlations were performed between the total scores of both the OASIS and the ODSIS, the DASS-14 and the EuroQoL. To provide useful clinical information, a Receiver Operating Curve (ROC) for each questionnaire was used to determine the optimum cut-off point that maximizes both sensitivity and specificity in depressive and anxiety symptomatology. The 75th percentile of the anxiety and depressive subscales of the DASS-14 were used as the validity criterion of the OASIS (DASS-Anxiety score = 5) and ODSIS (DASS-Depression score = 6), respectively. A similar threshold was suggested by previous studies on adolescents and young adults from the general population [53,54]. The questionnaires’ cut-offs were determined using the Youden Index, as per the following formula, where S represents the sensitivity and Sp the specificity:Y = S + Sp − 1

Lastly, the Information Functions of both questionnaires were calculated using the Samejima’s Graded Response Model within the IRT framework [55]. Data analyses were performed by means of FACTOR v10.10.03 [56], Mplus v8 [57], SPSS v26.0 [58] and IRTPRO v4.2 [59].

## 3. Results

### 3.1. Descriptive Data

The OASIS and ODSIS total scores had a mean of 3.92 (*SD* = 4.13) and 2.79 (*SD* = 4.06), respectively. Due to the Kolmogorov–Smirnov test’s sensitivity to larger sample sizes, normality was examined using skewness (*S*) and kurtosis (*K*) values lower than 2 and 7, respectively [60]. Despite some skewness, both total scores (OASIS: *S* = 1.12, *K* = 0.72; ODSIS: *S* = 1.39, *K* = 1.09) exhibited normal distributions. There were no significant differences between sexes in either the OASIS (males: *M* = 4.50, *SD* = 4.32; females: *M* = 3.81, *SD* = 4.09; *t*(375) = −1.159, *p* = 0.247) or ODSIS (males: *M* = 3.34, *SD* = 4.90; females: *M* = 2.91, *SD* = 3.90; *t*(67.70) = −0.618, *p* = 0.539).

### 3.2. Validity Evidence Based on OASIS Internal Structure

In the case of the OASIS, the Kaiser–Meyer–Olkin test (index = 0.870), the Barlett Sphericity test (χ^2^_(10)_ = 834.3, *p* < 10^−7^) and the goodness of fit indices (GFI = 0.998; RMSR = 0.045) suggested the adequacy of analysis and the model fit. The results also suggested a unidimensional structure (UniCo = 0.997; ECV = 0.937; MIREAL = 0.213) with 87.53% of explained variance (See Table 2). The CFA results in the second sample were not entirely satisfactory (χ^2^/df = 7.89, CFI = 0.937; RMSEA = 0.000, 95% Confidence Interval (CI) = 0.000–0.095). Thus, and with the aim of improving the model fit, the parameters from correlations between the errors of items 1 and 2 were released. The new results supported the above mentioned unidimensionality (χ^2^/df = 0.718, CFI = 0.999).

### 3.3. Validity Evidence Based on ODSIS Internal Structure

The Kaiser–Meyer–Olkin test (index = 0.894), the Barlett Sphericity test (χ^2^_(10)_ = 916.5, *p* < 10^−7^) and the goodness of fit indices (GFI = 0.999; RMSR = 0.028) also suggested the model adequacy for the ODSIS. Unidimensionality statistics suggested a unidimensional structure (UniCo = 0.996; ECV = 0.941; MIREAL = 0.190) with 90.60% of explained variance as indicated by the Parallel Analysis (See Table 3). The internal structure was cross-validated in the second subsample, as shown by the overall goodness of fit criteria from the CFA (χ^2^/df = 4.392, CFI = 0.955; but RMSEA = 0.134, 95%CI = 0.080–0.194).

### 3.4. Reliability

Both OASIS and ODSIS showed excellent reliability as estimated by the ordinal Cronbach’s alpha (α = 0.94 and 0.95, respectively) and McDonald’s omega (ω = 0.94 and 0.97, respectively). Discrimination indices of the OASIS items were over 0.83 (see Table 2) and those from the ODSIS were over 0.86 (see Table 3).

### 3.5. Validity Evidence Based on Relationships with Other Variables

Pearson zero-order correlations between the OASIS, the ODSIS, the DASS-14 and the VAS-EuroQoL are shown in Table 4. All questionnaires were significantly and positively correlated with the only exception of the VAS-EuroQoL, which was negatively correlated with the other measures. The OASIS and the ODSIS total scores presented associated variances with the respective DASS-14 subscales of 43.96% and 54.46%. In relation to quality of life, the associated variance of the OASIS was 11.42% and for the ODSIS was 16.08%.

Both OASIS and ODSIS offered a good discrimination of individuals reporting anxiety and depressive symptomatology, according to the ROC curve (See Figure 1). Specifically, the area under the curve for both questionnaires was 0.83 and 0.84, respectively. The total score that maximizes the sensitivity and specificity in both the OASIS (Sensitivity = 81.70%, Specificity = 73.80%) and ODSIS (Sensitivity = 69.20%, Specificity = 89.40%) was 4 and 5, respectively.

### 3.6. Item Response Theory Analysis

Figure 2 shows the items’ discrimination parameter α across levels of each variable (θ). As per the information functions, the highest precision in the OASIS is reached for values of θ between 0.5 and 1.5 (See Figure 2A). Regarding the ODSIS, the highest precision is reached between 0.5–2.5 (See Figure 2B).

## 4. Discussion

This study is the first exploring, in a nonclinical Spanish sample, the validity evidence of OASIS and ODSIS, and also the first time reporting evidence of validity in university students. The contributions made are innovative and necessary, bringing to light the scientific and clinical implications that the results exposed in this study may have.

In line with other studies [29,30], the internal structure suggested by the factor analysis supported a one-factor structure. Of note is that as the absence of anxiety symptoms necessarily implies a score of 0 in element 2 (intensity of symptoms), items 1 and 2 are correlated and the covariance between them should be considered when analyzing its internal structure [26,38].

The present study suggests a cut-off of 4 for the OASIS and 5 for the ODSIS (81.70% sensitivity for the OASIS, and 69.20% for the ODSIS). This is a novel finding due that none of the previous studies have performed this analysis in a sample of non-clinical university students. Although previous studies explored their validity evidence with nonclinical or student samples [29,30,32,34,36,43], none of them calculated the cut-off point. However, the present results can be compared in terms of mean scores. For example, the average OASIS score obtained in the present study (*M* = 3.92) is much closer to Ito and Oe et al.’s [32] result in a nonclinical sample (*M* = 3.67) than to the one reported by Norman et al. [29] in university students (*M* = 7.16). Regarding the ODSIS, present results (*M* = 2.79) are closer to those found by [30] in students (*M* = 2.57) than to those found by Ito et al. [36] in a nonclinical sample (*M* = 5.57). These differences may be due to the specific characteristics of the sample at the sociocultural level (e.g., nationality, sex, age), or to the recruitment process of the sample (e.g., online vs. paper-and-pencil).

The only study for which there are cut-off data in a non-clinical sample, specifically in a community population, is the one from the Czech Republic [34]. The high cut-off points suggested in this study are far from those of the present study (15 for the OASIS, and 12 for the ODSIS) and also beyond those obtained in clinical samples. The authors explain that it may be due to the characteristics of the sample and the pandemic process in which the questionnaires were administered.

We found Cronbach’s alphas of 0.94 for OASIS and 0.95 for ODSIS which demonstrate excellent internal consistency. These scores are within the range (0.80 ≤ α ≤ 0.96) observed in the literature [29,30], implying that the items were well validated and understandable by the participants in this study.

Although statistically significant correlations between the OASIS/ODSIS and other questionnaires included in the present study were in the hypothesized directions, it is worth mentioning the significant correlations between the OASIS and the DASS-14 depression subscale, between the ODSIS and the DASS-14 anxiety subscale, and between the OASIS and the ODSIS. These facts may be explained by the high comorbidity that the EDs present among themselves [38].

Finally, the highest precision of the OASIS and ODSIS was obtained for medium and moderately high levels of anxiety and depression symptoms. This result is of special interest, considering the nonclinical nature of the sample but the relatively high prevalence of ED symptoms among university students. Therefore, the IRT models suggest these instruments are very useful for research and clinical purposes when assessing emotional symptoms in this population.

This study presents several limitations. First, women represented 85.60% of the total sample, thus being sex distribution unbalanced. Although this could have influenced the results of the present investigation, it should be noted that there were no statistically significant differences in the OASIS and ODSIS scores based on sex. Additionally, EDs are more prevalent in females [2]. Second, the difficulty of evaluating whether the study participants actually had a clinical diagnosis or not through an online survey. To do this, we asked two questions in the evaluation battery: “In the past, have you received any kind of psychological treatment?” and “At present, are you receiving any kind of psychological treatment?” in order to be able to differentiate those participants who could be considered “clinical” from the “nonclinical”. However, it should not be forgotten that a person may be receiving psychological treatment and not present a clinical diagnosis or clinical symptoms as such. In future research, it would be interesting to be able to evaluate this variable more exhaustively, for example, administering an interview or writing questionnaires to evaluate the presence of symptoms and mental health disorders. Third, OASIS and ODSIS cut-offs were based on the DASS-14 scores rather than on established clinical criteria. Future studies ought to use alternative clinical criteria such as specific screening scores of diagnostics. Finally, the characteristics of the sample that make up this study are very specific, since they are university students, so the results should not be generalized to another type of sample or population.

## 5. Conclusions

The validation of the OASIS and the ODSIS in Spanish university students provides several advantages at research and clinical level. On the one hand, it allows us to have normative scores from Spanish samples, offering normative data (mean scores, standard deviations and Cronbach’s alpha) that can be compared with clinical ones to perform certain statistical analysis, such as the reliable change index [61]. On the other hand, and much more importantly, having two validated transdiagnostic instruments in a university sample allows us to improve the evaluation of transdiagnostic processes early and at a stage that involves many stressful events, facilitating the early detection of EDs in this population. Finally, both scales will facilitate the derivation of ED cases to the appropriate services in order to offer them efficient and fast care that improves their long-term prognosis.

## Figures and Tables

**Figure 1 ijerph-19-00345-f001:**
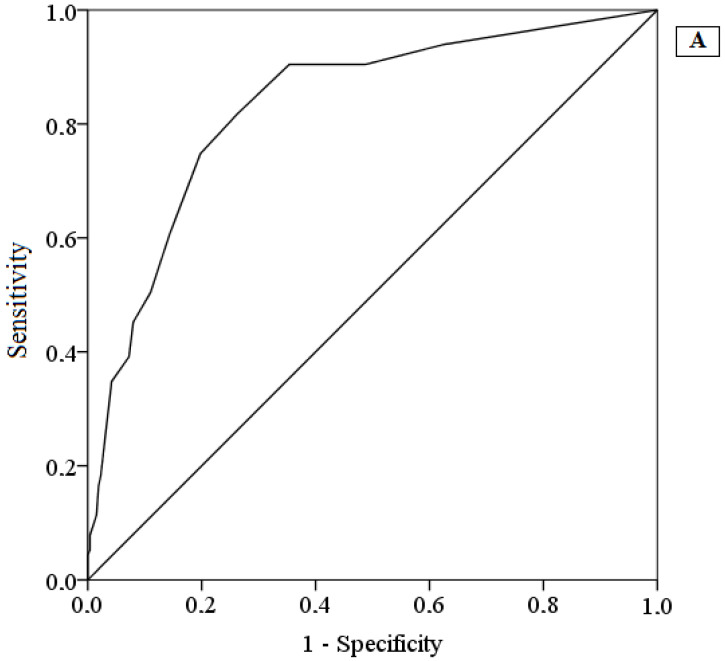
Receiver operating characteristic curve of the OASIS (**A**) and ODSIS (**B**).

**Figure 2 ijerph-19-00345-f002:**
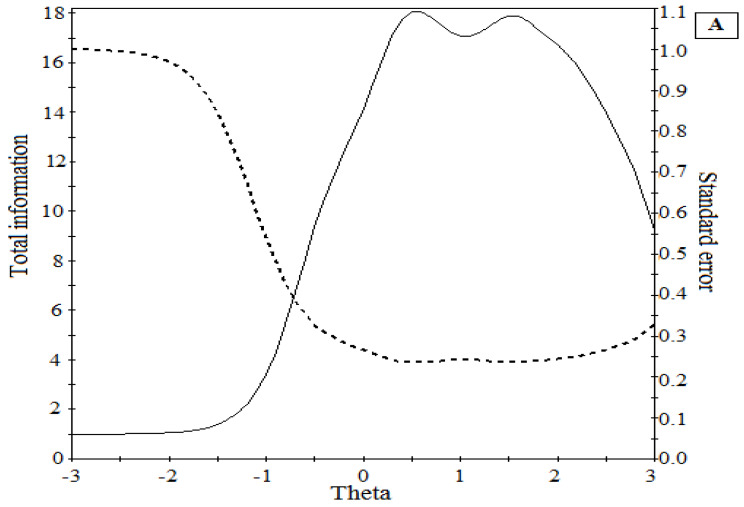
Information function of the OASIS (**A**) and ODSIS (**B**). The solid line represents the information function and the dotted line represents the standard error.

**Table 1 ijerph-19-00345-t001:** Socio-demographic characteristics of the participants.

Socio-Demographic Characteristics	*n*	%
Marital status		
Single	256	67.7
Married or with a partner	117	31
Divorced	5	1.3
Employment situation		
Not active (retired, unemployed, on sick leave)	256	67.7
Active	122	32.3
Ongoing university studies		
Psychology	211	55.8
Master in General Health Psychology	52	13.8
Doctorate	26	6.9
Nursing	19	5.0
Teacher training	14	3.7
Medicine	9	2.4
Other	47	12.4
Past psychological treatment		
No	252	66.7
Yes	126	33.3

**Table 2 ijerph-19-00345-t002:** OASIS factor loadings, discrimination indices and discrimination parameter *a*.

Items	Loading	DI	*a*
1.	0.893	0.808	4.15
2.	0.863	0.787	3.80
3.	0.832	0.721	2.77
4.	0.853	0.752	3.07
5.	0.888	0.797	3.90

DI: discrimination index items; *a*: IRT parameter *a*.

**Table 3 ijerph-19-00345-t003:** ODSIS factor loadings, discrimination indices and discrimination parameter *a*.

Items	Loading	DI	*a*
1.	0.859	0.802	3.85
2.	0.885	0.839	4.56
3.	0.888	0.810	4.01
4.	0.931	0.866	5.38
5.	0.865	0.820	4.29

DI: discrimination index items; *a*: IRT parameter *a*.

**Table 4 ijerph-19-00345-t004:** Correlations between OASIS, ODSIS, DASS-14-A, DASS-14-D and VAS-EuroQol.

	2	3	4	5
1. OASIS	0.653 *	0.663 *	0.599 *	−0.338 *
2. ODSIS	1	0.510 *	0.738 *	−0.401 *
3. DASS-14-A		1	0.616 *	−0.304 *
4. DASS-14-D			1	−0.381 *
5. VAS-EuroQol				1

OASIS: Overall Anxiety Severity and Impairment Scale; ODSIS: Overall Depression Severity and Impairment Scale; DASS-14-A: Anxiety subscale of the Depression, Anxiety and Stress Scales-14; DASS-14-D: Depression subscale of the Depression, Anxiety and Stress Scales-14; VAS-EuroQoL: Visual Analogue Scale of European version of the Quality of Life questionnaire. * *p* < 0.001.

## Data Availability

Data are available under request to the corresponding author.

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
