# Peer review of "Spanish Adaptation of the Overall Anxiety and Depression Severity and Impairment Scales in University Students"

_ijerph, 2021, doi:10.3390/ijerph19010345_

Round 1

Reviewer 1 Report

The aim of the study reported in this manuscript was to assess the validity of the Overall Anxiety/Depression Severity and Impairment Scales (OASIS and ODSIS, respectively) among Spanish university students. The manuscript is of clinical relevance, it is well written and provides important findings regarding the factor structure of these measures as well as its usefulness in clinical practice and research, particularly in a transdiagnostic framework. However, there are some issues that should be considered before proceeding with publication. Overall, I have some (generally) minor concerns about the paper in its present form, which I believe can be easily addressed by the authors. My specific comments and questions are detailed below.

Regarding the study rationale, I totally agree that the university period may be rather challenging for the mental health of students, and the authors state that clearly. However, it is also true that the presence of psychological symptoms or even ED may also impact the university experience (social and academic). Therefore, considering this potential bidirectional association, I suggest including some notes that this assessment is also important because the presence of mental health disorders may also impact (negatively) some university-related outcomes.

In the paragraph of line 50, I suggest support with a reference the second sentence regarding the transdiagnostic nature. It is not clear yet that these scales apprehend the transdiagnostic nature of ED. As well, in the last sentence, I suggest revising the sentence stating that these instruments are usually lengthy and time-consuming. Some of these may be, but not all (e.g., PHQ-9, which is mentioned in the references, is brief and easy to complete).

In Table 1 it is unclear what the authors mean by level of studies achieved. If the sample consisted of university students, how would you explain that about 30% have basic or mid-level education. This does not make sense and should be clarified.

It is particularly worrisome that the internal consistency of EuroQol is so low. The authors have any justification for this low value? This should be mentioned in the limitations.

In data analysis, please describe (even if briefly) the procedure to split the sample in two halves. Was this procedure random? These subsamples are equivalent in demographic variables? Some information regarding this procedure is important. The sum of these two subsamples slightly differs from the total sample size (in 4 participants) and it would be important to clarify this difference

In the results, I suggest providing the RMSEA (and CI) also as an index of model’s adjustment. As well, I would separate reliability and validity analyses (e.g., point 3.5 should be presented before point 3.4).

Please revise Table 4. There is something wrong with the table and the results are not possible to be accurately read (e.g., correlation between OASIS and ODSIS is not 1).

One important finding relates to the cut-offs of the OASIS and ODSIS. These cut-offs are particularly low (I assume the results range from 0 to 20) and lower than those reported in other studies in both clinical and non-clinical samples. In my opinion, this discussion should be expanded and strengthened. How would you explain this difference? May this be associated with the sample or the gold standard? The interpretation based on mean scores is not entirely convincing.

As well, the authors state that none of the previous studies calculated the cut-off point (line 244), however, in the introduction (lines 68-72), the authors mention some cut-offs within the same studies. How would you explain this incongruence? Can you clarify?

As the authors have the variable past psychological treatment, wouldn’t be interesting to explore if the OASIS and ODSIS differentiate individuals considering this variable (as a form of known-groups validity?). To some extent, it could be important so see if these two measures discriminate individuals with a potential vulnerability to psychopathological symptoms or to develop an ED.

Minor issues: (1) there are some typos in the manuscript that should be revised (e.g., line 16, live instead of life; line 50, related to, instead of related with); (2) in the abstract, I suggest removing the term “background” as it seems it is a unstructured abstract and the remaining subtitles are not present; (3) the authors do not need to present the internal consistency of the OASIS/ODSIS in the instruments description, as these are results of the study; (4) do not forget to remove the blind note in the final version; (5) some italics in p values or M are missing and should be revised.

Author Response

Dear reviewer,

We would like to thank you for your time and your valuable suggestions and comments who have helped us to significantly improve the manuscript. We have tried to provide an answer to each of them.

Below you can find the answer to each of your comments. We hope we have been able to respond properly to all of them.

We remain at your disposal for any further clarification.

The authors

Reviewer 1

The aim of the study reported in this manuscript was to assess the validity of the Overall Anxiety/Depression Severity and Impairment Scales (OASIS and ODSIS, respectively) among Spanish university students. The manuscript is of clinical relevance, it is well written and provides important findings regarding the factor structure of these measures as well as its usefulness in clinical practice and research, particularly in a transdiagnostic framework. However, there are some issues that should be considered before proceeding with publication. Overall, I have some (generally) minor concerns about the paper in its present form, which I believe can be easily addressed by the authors. My specific comments and questions are detailed below.

Regarding the study rationale, I totally agree that the university period may be rather challenging for the mental health of students, and the authors state that clearly. However, it is also true that the presence of psychological symptoms or even ED may also impact the university experience (social and academic). Therefore, considering this potential bidirectional association, I suggest including some notes that this assessment is also important because the presence of mental health disorders may also impact (negatively) some university-related outcomes.

Response: We appreciate the reviewer comment about the inclusion of the bidirectional association between the presence of mental health disorders and academic performance outcomes. Following this suggestion, we have included the following paragraph with its corresponding reference:

Likewise, it is important to consider that the presence of mental health disorders may be negatively associated and impact some university-related outcomes or academic functioning [14].”

In the paragraph of line 50, I suggest support with a reference the second sentence regarding the transdiagnostic nature. It is not clear yet that these scales apprehend the transdiagnostic nature of ED. As well, in the last sentence, I suggest revising the sentence stating that these instruments are usually lengthy and time-consuming. Some of these may be, but not all (e.g., PHQ-9, which is mentioned in the references, is brief and easy to complete).

Response: Following the reviewer’s recommendation, the wording of this paragraph has been amended to reflect the exact meaning of “transdiagnostic nature of these scales”. We have specified that “these scales are not designed to capture the severity and impairment of these disorders, regardless of the specific disorder suffered by the patients [26, 27]…” and we have supported the sentence with two references:

Regarding the second comment, the sentence starting that “these instruments are usually lengthy and time-consuming” has been softened and expressed in a different manner. We have also added a reference regarding the new statement:

In addition, the use of brief instruments to assess multiple anxiety disorders provides a more efficient way to collect data compared to longer scales [28], which may be less more useful in public clinical settings.”

In Table 1 it is unclear what the authors mean by level of studies achieved. If the sample consisted of university students, how would you explain that about 30% have basic or mid-level education. This does not make sense and should be clarified.

Response: Thank you so much for underlying this mistake. We have removed this variable as all participants were university students.

It is particularly worrisome that the internal consistency of EuroQol is so low. The authors have any justification for this low value? This should be mentioned in the limitations.

Response: We are very sorry for this mistake. From the EuroQol scale, we only use the visual analog scale (VAS), in which the participant must indicate their perceived state of heath on the continuous line from 0 (the worst imaginable state of health) to 100 (the best imaginable state of health). As it is a single item that collects quantitative information, we should not have calculated Cronbach's alpha. In this sense, it is necessary to eliminate this statistic, and specify in the instruments section that only the VAS has been used. We have also indicated that we will refer to this variable as VAS-EuroQol. This part of the EuroQol has also been used in other studies to assess self-perceived health status (e.j., Gili et al., 2020).

In data analysis, please describe (even if briefly) the procedure to split the sample in two halves. Was this procedure random? These subsamples are equivalent in demographic variables? Some information regarding this procedure is important. The sum of these two subsamples slightly differs from the total sample size (in 4 participants) and it would be important to clarify this difference

Response: We apologize for the lack of clarity on this point. Indeed, the total sample was randomly divided into two subsamples of the same size. As divided randomly, potential differences would be due to chance. We have also change the total sample size in the “participants” subsection to be consistent with the real sample size used.

In the results, I suggest providing the RMSEA (and CI) also as an index of model’s adjustment. As well, I would separate reliability and validity analyses (e.g., point 3.5 should be presented before point 3.4).

Response: We have now included the RMSEA and CI for both the OASIS and ODSIS. As now reported in the manuscript, the value for the OASIS is consistent with the other criteria but not for the ODSIS. Nonetheless, the CI included the value typically used as optimal. Thus, the overall valuation of the goodness of fit criteria in the cross-validation sample together with the EFA and the lasting analyses supports the structure.

Please revise Table 4. There is something wrong with the table and the results are not possible to be accurately read (e.g., correlation between OASIS and ODSIS is not 1).

Response: We apologize for this mistake. The data reported are correct but the correlation between OASIS and ODSIS appeared twice and with incorrect number in columns. We have now amended the error.

One important finding relates to the cut-offs of the OASIS and ODSIS. These cut-offs are particularly low (I assume the results range from 0 to 20) and lower than those reported in other studies in both clinical and non-clinical samples. In my opinion, this discussion should be expanded and strengthened. How would you explain this difference? May this be associated with the sample or the gold standard? The interpretation based on mean scores is not entirely convincing.

Response: Thank you very much for the appreciation. Thanks to this comment and the next one, we have realized that in the introduction it was not clear that, up to now, there are really no cut-off data for non-clinical samples. Based on this, we have specified the data available in this regard, to justify the comparison based on the mean scores and not based on the cut-off points.

As well, the authors state that none of the previous studies calculated the cut-off point (line 244), however, in the introduction (lines 68-72), the authors mention some cut-offs within the same studies. How would you explain this incongruence? Can you clarify?

Response: We are very sorry for this inconsistency between the introduction and the discussion. It is true that there are cut-off data in studies that use a sample made up of clinical and non-clinical participants, but actually, for the analysis of the ROC curve, they only use the subsample made up of clinical participants. Thanks to the suggestion made by the reviewer we have been able to clarify this in the introduction as well in the discussion section.

As the authors have the variable past psychological treatment, wouldn’t be interesting to explore if the OASIS and ODSIS differentiate individuals considering this variable (as a form of known-groups validity?). To some extent, it could be important so see if these two measures discriminate individuals with a potential vulnerability to psychopathological symptoms or to develop an ED.

Response: We thank the reviewer for this suggestion. However, it is important to note that participants comes from the general population and, therefore, they are not currently under psychological treatment. Thus and although in the past some underwent psychological treatment, results from the proposed analysis would not lead to any unquestionable conclusion, as both results (significant differences or non-significant differences) could be interpreted in terms of either lasting vulnerability or successfully recovery.

Minor issues: (1) there are some typos in the manuscript that should be revised (e.g., line 16, live instead of life; line 50, related to, instead of related with); (2) in the abstract, I suggest removing the term “background” as it seems it is a unstructured abstract and the remaining subtitles are not present; (3) the authors do not need to present the internal consistency of the OASIS/ODSIS in the instruments description, as these are results of the study; (4) do not forget to remove the blind note in the final version; (5) some italics in p values or M are missing and should be revised.

Response: We appreciate the reviewer’s corrections about some typos in the manuscript. We have corrected them within the manuscript.

Reviewer 2 Report

Thank you very much for allowing me to review the manuscript entitled “Spanish adaptation of the Overall Anxiety and depression Severity and Impairment Scales in University students .” This study examined the reliability, factorial validity, convergent and discriminant validity of OASIS and ODSIS among the nonclinical university students in Spain. In addition, this study reported the cut-offs and precision information for OASIS and ODSIS. The manuscript was written well. The analyses conducted were adequate except for the ROC analysis. The results presented in this study were important because there were insufficient findings among nonclinical university students. As the authors argue, the transdiagnostic screening instrument is important to mental health conditions in nonclinical populations. I have two major and two minor comments.

[Major Points]

ROC analysis is usually used to assess the screening performance using diagnostic status assessed by more rigorous methods (e.g., SCID) or actual clinical states (e.g., having outpatient treatment). This study used DASS as such criterion. However, DASS is just another type of self-report measure for anxiety and depression. Hence, it seems less persuasive to conduct ROC analysis for assessing the screening performance of OASIS and ODSIS. AS written in the limitation section, the authors seem to understand this aspect. If authors decide to retain this analysis, it would be beneficial for readers to explain more extensively about the evidence using the cut-off scores of DASS in the Method section.

I guess Table 4 was mispresented. Correlations between OASIS and 1 should be 1. Correlations between OASIS and 2(ODSIS) should be .653. In this way, the authors need to rearrange the figures.

[Minor Points]

In the Abstract, “quality of live” should be “quality of life.”

There is no need to report Cronbach alpha for OASIS and ODSIS in Method section because it is presented in the Result section.

Author Response

Dear reviewer,

We would like to thank you for your time and your valuable suggestions and comments who have helped us to significantly improve the manuscript. We have tried to provide an answer to each of them.

Below you can find the answer to each of your comments. We hope we have been able to respond properly to all of them.

We remain at your disposal for any further clarification.

The authors

Reviewer 2

Thank you very much for allowing me to review the manuscript entitled “Spanish adaptation of the Overall Anxiety and depression Severity and Impairment Scales in University students.” This study examined the reliability, factorial validity, convergent and discriminant validity of OASIS and ODSIS among the nonclinical university students in Spain. In addition, this study reported the cut-offs and precision information for OASIS and ODSIS. The manuscript was written well. The analyses conducted were adequate except for the ROC analysis. The results presented in this study were important because there were insufficient findings among nonclinical university students. As the authors argue, the transdiagnostic screening instrument is important to mental health conditions in nonclinical populations. I have two major and two minor comments.

 [Major Points]

ROC analysis is usually used to assess the screening performance using diagnostic status assessed by more rigorous methods (e.g., SCID) or actual clinical states (e.g., having outpatient treatment). This study used DASS as such criterion. However, DASS is just another type of self-report measure for anxiety and depression. Hence, it seems less persuasive to conduct ROC analysis for assessing the screening performance of OASIS and ODSIS. AS written in the limitation section, the authors seem to understand this aspect. If authors decide to retain this analysis, it would be beneficial for readers to explain more extensively about the evidence using the cut-off scores of DASS in the Method section.

Response: We agree with the reviewer that validity criteria usually is explored using a more rigorous method than a self-reported measure. Nonetheless, it is worthy to mention that diagnostics rely also in self-reported information. Besides, the DASS-21 is a widely used instrument characterized by sound psychometric properties, tested against diagnostics criteria. It is important to note that the sample is comprised by young adults from the general population and thus, their symptomatology is, by definition, below clinical threshold.

As requested by the reviewer we have now elaborated more on the evidence of using the proposed cut-offs. Specifically, in addition to use a score obtained only by the upper 25% of the sample, previous studies on young adults have proposed virtually the same cut-offs for both adolescents and young adults from the general (Evans, Haeberlein, Chang, & Handal, 2020; Roman, Santibañez, & Vinet, 2016).

I guess Table 4 was mispresented. Correlations between OASIS and 1 should be 1. Correlations between OASIS and 2(ODSIS) should be .653. In this way, the authors need to rearrange the figures.

Response: We apologize for this mistake. The data reported are correct but the correlation between OASIS and ODSIS appeared twice and with incorrect number in columns. We have now amended the error.

 [Minor Points]

In the Abstract, “quality of live” should be “quality of life.”

Response: We appreciate the reviewer’s correction about this typo. We have corrected it within the abstract.

There is no need to report Cronbach alpha for OASIS and ODSIS in Method section because it is presented in the Result section.

Response: We have removed as suggested by the reviewer.

This manuscript is a resubmission of an earlier submission. The following is a list of the peer review reports and author responses from that submission.